# Epstein-Barr Virus-Associated Post-Transplant Lymphoproliferative Disorders after Hematopoietic Stem Cell Transplantation: Pathogenesis, Risk Factors and Clinical Outcomes

**DOI:** 10.3390/cancers12020328

**Published:** 2020-02-01

**Authors:** Ayumi Fujimoto, Ritsuro Suzuki

**Affiliations:** Department of Oncology and Hematology, Shimane University Hospital, Izumo 693-8501, Japan; fujimoto613033@gmail.com

**Keywords:** post-transplant lymphoproliferative disorder, hematopoietic stem cell transplantation, pathogenesis, risk factors

## Abstract

Epstein-Barr virus (EBV) is a ubiquitous virus belonging to the human γ-herpes virus subfamily. After primary infection, EBV maintains a life-long latent infection. A major concern is that EBV can cause a diverse range of neoplasms and autoimmune diseases. In addition, patients undergoing hematopoietic stem cell transplantation or solid organ transplantation can experience post-transplant lymphoproliferative disorders (PTLDs) due to dysfunction or suppression of host’s immune system, or uncontrolled proliferation of EBV-infected cells. In recent years, the number of EBV-associated PTLD cases has increased. This review focuses on the current understandings of EBV-associated PTLD pathogenesis, as well as the risk factors and clinical outcomes for patients after allogeneic stem cell transplantation.

## 1. Introduction

Epstein-Barr virus (EBV) infects more than 90% of the adult population worldwide at some point in their lives, usually with no ill effects [1]. EBV was first identified in 1964 from a patient with Burkitt’s lymphoma, suggesting that EBV is a causative agent of human cancer [2]. Since then, EBV has been identified as the cause of several human cancers, including nasopharyngeal carcinoma and Hodgkin’s lymphoma; it is also responsible for post-transplant lymphoproliferative disorder (PTLD) [3]. In 1969, Penn et al. first reported PTLD in five patients who developed malignant lymphoma after kidney transplantation [4]. Later, the term “PTLD” was introduced by Starzl et al. in 1984 [5]. PTLD is recognized as a life-threatening complication after transplantation [4,5]. After an initial infection, EBV maintains a life-long latent infection of memory B-cells; thus, the virus can cause a range of neoplasms attributable to dysregulated proliferation of EBV-infected B-cells due to dysfunction or suppression of the host immune system after transplantation. Therefore, the pathological manifestations of PTLD are heterogenous. The 2017 revised 4th edition of the World Health Organization classification recognizes four different entities: non-destructive PTLD characterized histologically by a lack of architectural effacement (plasmacytic hyperplasia, infectious mononucleosis-like PTLD, and florid follicular hyperplasia), polymorphic PTLD characterized by a full spectrum of lymphoid maturation but not satisfying the criteria for lymphoma, monomorphic PTLD (B-cell neoplasms and T/NK-cell neoplasms which are classified in more detail according to the historical characteristics of the lymphoma they most resemble), and classical Hodgkin lymphoma PTLD (Table 1). The most common histological subtype of monomorphic PTLD is diffuse large B-cell lymphoma, which accounts for ~60% of cases. Other subtypes such as Burkitt lymphoma, plasma cell neoplasms and T-cell lymphoma have been reported [6,7]. Most of PTLD cases are associated with EBV infection and subsequent oncogenesis (see below), although 10–48% of monomorphic PTLDs are EBV-negative [8].

## 2. Pathogenesis

### 2.1. EBV Infection and Latent Status

Initially, EBV infects naïve B-cells. EBV-positive naïve B-cells migrate to germinal centers in lymph nodes, mucosa-associated lymphoid tissue, or spleen. In germinal centers, normal B-cells undergo activation-induced cytidine deaminase-driven somatic hypermutation and class switch recombination of the antigen-binding variable region of immunoglobulin genes to increase the specificity of B-cell antibodies; affinity-based selection of B-cells occurs before maturation into plasma cells or memory B-cells [9]. The classical model of EBV infection is based on the finding that expression of EBV proteins by EBV-infected naïve B-cells gives them a selective advantage in the germinal center; these proteins also stimulate maturation into memory B-cells, which are the presumed reservoir of EBV, before final establishment of a latent EBV infection [10].

The life cycle of EBV is either latent state or lytic state [11]. Upon primary infection or reactivation of a latent infection, EBV runs a transient lytic program. In the lytic state, EBV DNA is replicated once at the S-phase, with synchronization of the host genome; this generates progeny viruses. EBV-infected cells express nearly 100 viral genes during replication [1]. However, EBV usually settles into a latent state referred to as latency 0. In this state, EBV genomic DNA resides in the nucleus as a ring-shaped episome that can integrate into the host genome. EBV-infected cells express only a few latent viral genes that allow the virus to persist for long periods during latency 0 [12]. Based on the expression pattern of EBV proteins including six types of EBV nuclear proteins (EBV nuclear antigen (EBNA) 1, 2, 3A–C and EBNA-leader protein) and three types of latent membrane proteins [latent membrane protein (LMP) 1, 2A,B), three different latency expression profiles in addition to latency 0 are recognized (Table 2) [13]. These different latency profiles are associated with different stages of EBV-infected B-cells and with different immune conditions. 

EBV-positive PTLD typically results from a latency III program, referred to as “growth program”, in which all nine viral proteins are expressed [14]. Typically, latency III is observed during EBV primary infection and in immunocompromised patients with lymphoma. By contrast, latency II, referred to as “default program”, is characterized by expression of LMP1, and a lack of EBNA2 and EBNA3. The number of EBV proteins expressed at this stage is more limited than that expressed during latency III, thereby minimizing the immunogenicity of infected cells to allow the virus to escape surveillance by cytotoxic T-cells. Latency I is characterized by very limited expression of EBV proteins; only EBV-encoded small RNA (EBER) 1 and 2 and EBNA1 are expressed. Rare PTLD subtypes presenting histologically as Burkitt lymphoma or plasma cell neoplasms almost always show a latency I pattern, whereas classical Hodgkin lymphoma PTLD presents with latency II pattern; overall, though, PTLD typically presents with a latency III pattern.

### 2.2. EBV-Induced Oncogenesis

Three important factors contributing to the pathogenesis of PTLD by EBV infection were suggested as follows: (1) EBV-encoded oncogenes, (2) host immune suppression, and (3) genetic or epigenetic alternations in the host [15]. The EBNA1 protein, which is expressed in all latency patterns, binds to EBV DNA to ensure EBV genomic replication; the protein resides in the nucleus of an infected B-cell as a circular DNA episome [1]. LMP1 and LMP2, which are expressed during latency II and III, act mainly as oncogenic proteins. These proteins mimic the B-cell surface molecule CD40 and the B-cell receptor, respectively, thereby activating several downstream signaling pathways, including the nuclear factor (NF)-κB and phosphatidylinositol-3 kinase/Akt pathways, which drive proliferation of EBV-infected naïve B-cells and guide them throughout the germinal center reaction, ultimately pushing the infected B-cells toward the memory B-cell stage in which EBV can persist [3,16,17,18,19]. During latency III, further expression of EBNA2 acts as a strong transcriptional coactivator for the LMP1 and LMP2 promoters, as well as the C promoter, further driving growth and transformation of EBV-infected B-cells. Although EBNA3, expressed only during latency III, is a target for cytotoxic T-cells, the number of cytotoxic T-cells in immunocompromised patients including patients after transplantation, is usually reduced by the conditioning regimen, and their function is also impaired by immunosuppressive agents; therefore, EBV-infected cells with a latency III pattern proliferate only under condition of immunosuppression. EBER 1 and EBER 2 as well as EBNA1 are expressed during all latency patterns; although they are the most abundantly expressed viral products in infected cells, their function is still unclear. 

### 2.3. Hematopoietic Stem Cell Transplantation Setting

Patients undergoing hematopoietic stem cell transplantation (HSCT) have reduced numbers of EBV-specific cytotoxic T-cells and impaired T-cell mediated immunity due to pre-transplant conditioning regimen and immunosuppressive agents. This allows proliferation of EBV-infected B-cells. The extended lifespan of these cells further allows acquisition of several genetic or epigenetic aberrations, including alterations to c-MYC, BCL6, and p53; microsatellite instability; and DNA hypermethylation [20]. In addition to the immunosuppressive environment, persistent immune activation and chronic inflammation contribute to the development of PTLD [21]. Pathogen-associated molecular patterns (PAMPs), which are structural components belonging to bacteria, fungi, and viruses (e.g., lipopolysaccharide, 16S ribosomal DNA, and CpG DNA) bind to Toll-like receptors (TLRs) and activate the innate immune system. Endogenous damage-associated molecular patterns (e.g., mitochondrial DNA, high mobility group box 1 protein, and defensins) are released by damaged cells; these also activate the immune system by binding to TLRs. PAMPs and damage-associated molecular patterns initiate a complex signal transduction cascade by binding to the extra- and intra-cellular domains of TLRs; this amplifies the TLR-mediated immune response and leads ultimately to increased transcription of pro-inflammatory cytokines such as interleukin (IL)-6 and tumor necrosis factor (TNF)-α. These pro-inflammatory cytokines cause chronic inflammation and drive proliferation of polyclonal EBV-infected cells. In the HSCT setting, the conditioning regimen (e.g., high dose chemotherapy and total body irradiation) often causes damage to the intestinal mucosa, thereby inducing release of pro-inflammatory cytokines such as TNF-α, type 1 interferons, IL-1, and IL-6. PAMPs are also released by the intestinal microbiota in response to the conditioning regimen and by other pathogens that may have infected the patient [22]. After neutrophil engraftment, damaged intestinal and other host tissues release the inflammatory cytokines such as TNF-α, IL-1, and lipopolysaccharide, which activate donor-derived T-cells; this triggers a “cytokine storm”, known as acute graft-versus-host disease (GVHD) [23]. Therefore, immunosuppression due to T-cell depletion and T-cell dysfunction, along with release of inflammatory cytokines caused by the conditioning regimen, provide conditions that are optimal for development of PTLD, particularly during the early phase post-HSCT (Figure 1).

### 2.4. Genetic or Epigenetic Alternations

Several genetic studies using different methods revealed that various chromosomal and genetic alterations were associated with PTLD, suggesting that EBV infection alone does not account for post-transplant lymphomagenesis [24,25,26,27]. However, the heterogeneity of PTLD and differences in analysis methods used meant that these studies yielded conflicting results. To date, several cytogenetic analyses of PTLD have been performed. One study that examined 36 PTLD cases, including 2 early lesions, 13 polymorphic PTLDs, and 21 monomorphic PTLDs (18 B-cell neoplasms and 3 T-cell neoplasms), showed that 72% of monomorphic B-cell PTLDs and all T-cell PTLDs contained chromosomal abnormalities, in contrast that only 15% of polymorphic PTLDs and none of the early lesion PTLDs did. The most common abnormality in monomorphic PTLD was trisomy 9 and/or trisomy 11, followed by translocations involving 8q24.1, 3q27, and 14q32 [25].

Interestingly, recent molecular genomic studies of post-transplant diffuse large B-cell lymphoma (PT-DLBCL) revealed that EBV-positive PT-DLBCL and EBV-negative PT-DLBCL have distinct genomic profiles [25,28,29,30]. Typically, EBV-positive PT-DLBCL occurs within 1 year after transplantation; therefore, it harbors fewer genomic abnormalities than EBV-negative PT-DLBCL or de novo DLBCL in immunocompetent patients. By contrast, EBV-negative PT-DLBCL typically occurs in late phase after transplantation and harbors at least 10 genomic aberrations recurrent in de novo DLBCL. These findings were validated by another copy number alteration analysis [28].

The most common copy number aberration in EBV-positive PT-DLBCL is the gain/amplification of 9p24.1 targeting PDCD1LG/PDL2. Gain of 9p24.1, a well-known aberration in primary mediastinal B-cell lymphoma, classical Hodgkin lymphoma and primary central nervous system (CNS) lymphoma, increases expression of PDL1, PDL2, and JAK2 protein of tumor cells, resulting in an escape from T-cell immunity and increased cell growth [31]. Interestingly, LMP1, an EBV-encoded protein expressed during latency II and III, upregulates the expression of PDL1 and contributes to tumor cells survival [32]. By contrast, common copy number aberrations in EBV-negative PT-DLBCL include gain of 3/3q and 18q, loss of 6q23/TNFAIP3, and loss of 9p21/CDKN2A [28]. Some of these results are consistent with previous findings [24]. Gain of chromosome 3/3q is unique to EBV-negative PT-DLBCL and is associated with a differential expression of various genes, including FOXP1. FOXP1 encodes a transcriptional regulator, and acts as both an oncogene and a tumor suppressor, which is associated with development of several types of cancer [33]. With respect to pathogenesis of non-Hodgkin lymphoma, gain of 3/3q is an unfavorable genetic aberration in activated B-cell DLBCL; FOXP1 expression is also documented in de novo DLBCL [34,35]. CDKN2A, which encodes cyclin-dependent kinase inhibitor 2A (p16^INK4a^), plays an important role in controlling cell growth by arresting the cell cycle at G1 [36,37]. Loss of CDKN2A is also an unfavorable genetic aberration in de novo DLBCL, along with the loss of TP53 [38]. Therefore, these genetic aberrations play an important role in the pathogenesis of EBV-negative PT-DLBCL, as well as de novo DLBCL.

A recent study performed targeted next generation sequencing of 68 genes to identify differences in somatic mutation profiles between EBV-positive and EBV-negative PT-DLBCL [29]. Compared with de novo DLBCL in immunocompetent patients, EBV-positive PT-DLBCL harbors fewer mutated genes, particularly genes associated with the NF-κB pathway. Although *TP53* mutations were more common in EBV-negative PT-DLBCL than in EBV-positive PT-DLBCL and de novo DLBCL, the overall mutational frequency, including gene clusters related to the NF-κB pathway and epigenetic modifiers, in EBV-negative PT-DLBCL was similar to that in de novo DLBCL.

In addition to genetic aberrations, epigenetic alterations are potentially associated with the pathogenesis of PTLD. The LMP1 oncogene induces cluster changes in the DNA methylation status of cellular genes depending on the CpG content of the promoter region by downregulating *DNMT1* and *DNMT3B*, and upregulating *DNMT3A* in germinal center B-cells [39]. Besides, death-associated protein kinase, O6-methylguanine-DNA methyl-transferase, *TP73*, *CDKN2A/INK4A*, and *PTPN6/SHP1* are hypermethylated, particularly in a part of monomorphic PTLD [40].

Taking into account all of the above, EBV-negative PTLD might be considered as a type of lymphoma that develops coincidentally in transplant recipients, although it is usually difficult to distinguish from treatment-related DLBCL. Other studies speculate that EBV-negative PTLD may develop after infection by Human Herpes virus 8 and cytomegalovirus, after chronic antigen stimulation by the graft, or after hit-and-run EBV infection, resulting in accumulation of genetic or epigenetic aberrations, and providing a particular tumor micro environment that promotes lymphomagenesis [41,42,43]. 

## 3. Epidemiology

The incidence of PTLD differs according to the type of transplanted organs. The incidence of PTLD after HSCT is lower than that after solid organ transplantation (SOT) (Table 3). PTLD is a common secondary malignancy after SOT, and the most common one is a non-melanoma skin cancer. The incidence is estimated to be 1–33%, with the highest incidence occurring in recipients of multi-visceral and intestinal transplants who receive higher amounts of immunosuppressive agents (7–33%), followed by recipients of lung transplants (3–10%), and heart transplants (2–8%); the lowest incidence occurs in recipients of kidney, pancreatic, or liver transplants (1–2%) [44,45,46,47]. Patients who receive SOT require life-long immunosuppressive agents, therefore, PTLD can occur in the late phase after SOT. The median onset of PTLD after SOT is significantly later than that of PTLD after HSCT, although the highest rate of PTLD incidence after SOT is seen in the first year post- transplantation [47,48]. The median time of onset post-transplantation is 4–5.3 years [6,48]. Of the PTLD cases that develop after SOT, most are of recipient origin [49]. Some donor-derived PTLD cases developed after SOT were reported, but they were commonly limited to allograft tissues [50]. By contrast, the incidence of PTLD after HSCT is approximately 0.8–4.0%, which is much lower than that after SOT, although the reported incidence ranges from 1% to 17% depending on patient characteristics, stem cell source, degree of HLA mismatch, and conditioning regimen [51,52,53,54,55,56,57,58,59,60,61,62]. Patients who received cord blood (CB) transplantation has higher risk of PTLD development than those who received bone marrow or peripheral blood stem cell transplantation, and the incidence of PTLD is 2.0–4.5% [63,64,65,66]. Because the patients after HSCT often stop taking immunosuppressive agents, thereby allowing reconstitution of EBV-specific T-cell mediated immunity within 6 to 12 months post-HSCT, PTLD typically develops within 1 year, whereas late-onset PTLD is rare. PTLD cases after HSCT are much frequently of donor origin [67,68,69]. The incidence of PTLD has increased over the past two decades, alongside an increasing number of HSCT particularly haploidentical HSCT, the introduction of new immunosuppressive agents and regimens, older age of donors and recipients, greater awareness of PTLD, and improved accuracy of PTLD diagnosis [61,62,70].

## 4. Risk Factors

There are several known risk factors for PTLD; these depend principally on the degree of T-cell depletion or dysfunction. The risk factors associated with PTLD after allogeneic HSCT are shown in Table 4. The most common risk factors are T-cell depletion strategies and donors other than HLA-matched related donors.

Owing to the increased number of allogeneic HSCTs from HLA-mismatched or unrelated donors, T-cell depletion strategies are also increasingly used as a conditioning regimen. Such strategies include in vivo depletion of T-cells using antithymocyte globulin (ATG) and ex vivo depletion by elutriation/density gradient centrifugation. The aim of these strategies is to reduce the risk of graft rejection and to reduce the risk of severe GVHD. T-cell depletion also removes EBV-specific cytotoxic T-cells; this procedure compromises T-cell mediated immunity, thereby increasing the risk of EBV reactivation and development of PTLD. Rabbit ATG is much more likely to cause profound lymphocytopenia than horse ATG [71]. Several studies show that T-cell depletion increases the risk of PTLD [52,53,58,62,72]. Landgren et al. indicated that selective T-cell depletion methods such as anti-T and anti-NK cell monoclonal antibodies (relative risk (RR) = 8.4), sheep red blood cell 8 rosetting (RR = 14.6), and lectin with/without sheep red blood cells or an anti-T monoclonal antibodies (RR = 15.8) increase the risk of PTLD to a greater extent than broad lymphocyte depletion methods such as alemtuzumab monoclonal antibody (RR = 3.1), or elutriation/density gradient centrifugation (RR = 3.2) [58]. In addition, we found that high dose ATG, defined as a total dose of thymoglobulin >2.5 mg/kg or ATG-F > 5.0mg/kg, was associated with a 2.3-fold higher risk of PTLD than low dose ATG, suggesting that ATG increases the risk of PTLD in a dose-dependent manner [62].

The degree of HLA matching is associated with development of PTLD. Uhlin et al. showed that the use of an HLA-mismatched donor (RR = 5.9) was associated with a higher risk of PTLD than the use of an HLA identical donor [61]. Another study indicated that the risk of PTLD depended on the degree of HLA mismatch: a related donor with two or more HLA antigen-mismatches (RR = 3.1) or an unrelated donor (RR = 4.2) significantly increased the risk of PTLD when compared with an HLA identical sibling donor, but a related donor with a single antigen-mismatch did not (RR = 1.8) [58]. Styczynski et al. demonstrated that the overall incidence of PTLD for a matched related donor was 1.16%, compared with 2.86% for a mismatched related donor, 3.97% for a matched unrelated donor, and 11.24% for a mismatched unrelated donor [60]. Interestingly, CB was associated with the greater risk of PTLD [62]. CB is associated with a 1.5- to 2.0-fold increased risk of PTLD when compared with an HLA-mismatched or unrelated donor. According to previous reports, evaluating the incidence of PTLD among CB recipients, the incidence of PTLD after CB transplantation is around 2.0–4.5% [63,64,65,66]. Low numbers of infused donor T-cells, T-cell naivety, or delayed antigen-specific cellular immune reconstitution during the early phase after HSCT may contribute to the high incidence of PTLD after CB transplantation [73]. Haploidentical allogeneic HSCT with post-transplant cyclophosphamide (PTCy) was introduced recently, and the number of this procedure is increasing. The incidence of PTLD after haploidentical HSCT is unexpectedly low at 0–3.0% [74,75]. Previous studies also report that PTLD does not develop after haploidentical HSCT with PTCy [76,77,78]. Possible reasons for the relatively low incidence of PTLD after haploidentical HSCT with PYCy include destruction of donor and recipient EBV-infected B-cells, relative sparing of EBV-specific memory T-cells, and more rapid T-cell immune reconstitution than occurs after ATG use; however, the data are still limited [79].

Although various other risk factors have been reported, they differ according to the patient characteristics, conditioning regimen, and immunosuppressive agents used; thus, their impact on development of PTLD is less clear. The use of reduced intensity conditioning regimens is increasing, along with the number of HSCT procedures performed in elderly patients. However, studies show that reduced intensity conditioning regimens delay reconstitution of EBV-specific immunity, thereby increasing the risk of PTLD (RR = 3.3) [61,81]. GVHD and immunosuppressive agents also delay T-cell immune reconstitution [82]. GVHD impairs T-cell functions by limiting T-cell receptor diversity, and T-cell development during the pro-inflammatory cytokine storm [83]. Several studies showed that acute GVHD increases the risk of PTLD (RR = 1.7–2.7) [58,61,62]. EBV serological mismatch, particularly the combination of a serologically EBV-negative recipient and a serologically EBV-positive donor, is also reported as a risk factor in HSCT patients [54,61,80]. EBV-negative recipients lack EBV-specific cytotoxic T-cells; thus if they receive HSCT from an EBV-positive donor, then the donor-derived EBV-infected B-cells flourish in an environment that lack EBV-specific T-cell mediated immunity, resulting in PTLD. Regarding primary diseases of patients, aplastic anemia, primary immunodeficiency disease, chronic myeloid leukemia, and advanced Hodgkin’s lymphoma increase the risk of PTLD [53,62,70]. Reactivation of cytomegalovirus is also strongly associated with EBV reactivation and PTLD development because patients with reactivated cytomegalovirus may be under severe immunosuppression, placing them at high risk of infection by other viruses [84].

Previous studies suggested that various risk classifications to identify high risk patients who may benefit from early intervention. The risk factors used for each classification are different among studies, and include both pre-transplant and post-transplant parameters (Table 5). Based on the large database of 26,901 patients after HSCT collected from the Center for International Blood and Marrow Transplant Research and the Fred Hutchinson Cancer Center, Landgren advocated a risk predictive model according to the sum of four major risk factors: selective T-cell depletion methods, ATG use for GVHD prophylaxis or treatment, two HLA antigen mismatched or unrelated donors accompanied by selective T-cell depletion, and age 50 years or older [58]. The cumulative incidence of PTLD was estimated as 0.2%, 1.1%, 3.6%, and 8.1%, based on 0, 1, 2, and 3–4 risk factors, respectively. Another risk classification created by Karolinska University Hospital included seven risk factors listed in Table 5. 

Incidence of PTLD was estimated as 0.4%, 3.0%, 10.4%, 26.5%, and 40%, based on 0–1, 2, 3, 4, and 5 of the seven risk factors, respectively [61]. This classification is based on a database from single center. Therefore, this model includes detailed patient information such as EBV infection status between recipient and donor and mesenchymal stromal cell treatment for GVHD. Recently, a novel 5-point scoring system was developed based on Japanese registry database. This scoring gave different weights to each risk factor and was based only on pre-transplant risk factors: ATG used in the conditioning regimen (high dose, 2 points; low dose, 1 point); donor type (HLA-mismatched related donor, 1 point; unrelated donor, 1 point; CB, 2 points), and primary disease (aplastic anemia, 1 point) [62]. The points are summed and patients are classified into four risk groups according to the estimated incidence of PTLD at 2 years after HSCT: low risk (0–1 point), probability 0.3%; intermediate risk (2 points), probability 1.3%; high risk (3 points), probability 4.6%; very high risk (4–5 points), probability 11.5% (Figure 2). These scoring systems are useful for estimating the risk of PTLD before allogeneic HSCT, although all require further validation.

## 5. Clinical Presentation

Typically, PTLD after HSCT develops within 1 year, before the reconstitution of EBV-specific cytotoxic T-cell immunity [52,54]. Thus, late-onset PTLD is much less common after HSCT than after SOT [85,86,87,88]. It is documented that EBV-negative PTLDs tend to occur during the late phase after transplantation. However, most reports analyzed EBV-negative PTLD cases occurring in SOT recipients. Although EBV-negative PTLDs occur significantly later (median onset 4–5 years) than EBV-positive PTLDs in patients after SOT, the onset time in terms of EBV positivity is not different among those after HSCT (EBV-negative cases: median onset 5 months) [86,87,88,89]. With respect to allografts, analysis of our previous data suggested that the median onset days of PTLD development was later in patients who received CB transplantation (202 days) than in those who received bone marrow or peripheral blood stem cell transplantation (111 days) [62]. This might be attributed to a low number of infused T-cells in the CB graft, and to delayed antigen-specific immune reconstitution after CB transplantation. The clinical manifestations of patients with PTLD are highly variable depending on the morphologically defined category of PTLD, localization of PTLD, and the patient’s general condition. Fever and lymphadenopathy are the most common symptoms, although some PTLDs develop with nonspecific symptoms such as prolonged fever, sweats, general malaise, and weight loss, and others are found incidentally. By contrast, some PTLDs show common symptoms of malignant lymphoma such as lymphadenopathy, swelling of tonsils or adenoids, and hepatosplenomegaly. As it progresses, PTLD can involve any organ, including bone marrow, liver, spleen, lung, gastrointestinal tract, and kidney, even the CNS in some cases (Figure 3). Thus, PTLD may present with organ-specific symptoms such as abdominal pain, gastrointestinal bleeding, or dyspnea [55,57,59,90]. PTLD after HSCT often progresses rapidly, and Ann-Arbor advanced stage of PTLD is more common in patients after HSCT than in those after SOT [48]. As a rare presentation, disseminated PTLD can sometimes present like fulminant sepsis or severe GVHD [91]. Regarding laboratory tests, the number of EBV-DNA copies in peripheral blood and lactate dehydrogenase levels in serum increase progressively. In cases with organ involvement, laboratory data such as liver enzymes or kidney tests can be elevated. Differential diagnoses include GVHD, hemolytic anemia, toxoplasma, tuberculosis, and other virus infections such as cytomegalovirus, varicella zoster virus, adenovirus, or hepatitis B virus, which can co-occur with PTLD [92]. ^18^F-FDG-PET/CT has high sensitivity for PTLD and is useful for detecting disease lesions [93,94]. Because PTLD is usually FDG-avid, the Lugano classification by PET-CT is recommended for the staging of PTLD [95,96]. For precise diagnosis of PTLD, a surgical biopsy of suspicious lesions with the highest FDG uptake, is desirable. Measurement of EBV copy number in the peripheral blood using polymerase chain reaction is also important and helpful for diagnosis of EBV-positive PTLD. However, although the detection of EBV-DNA is highly sensitive, it has low positive predictive value for PTLD. If the biopsy is not easy, a combination of non-invasive approaches including ^18^F-FDG-PET/CT and measurement of EBV DNA can be considered for early diagnosis and/or treatment.

## 6. Treatments

The ECIL-6 guideline of PTLD classifies management strategies for PTLD into three categories: prophylaxis, pre-emptive therapy and targeted therapy [97]. Treatments for PTLD comprise reduction of immunosuppression (RI), rituximab, chemotherapy, and adoptive immunotherapy. Few studies have evaluated the different treatments for PTLD in the setting of HSCT due to its rare incidence and heterogeneity.

### 6.1. Propylaxis

Prophylaxis involves intervention to prevent EBV DNAemia in asymptomatic EBV-seropositive patients. However, a standard of prophylaxis for EBV DNAemia is not established. Rituximab, a monoclonal anti-CD20 antibody, is effective for prophylactic therapy against EBV reactivation that reduces the risk of, and mortality from, PTLD development, particularly in high-risk patients [98,99,100]. A retrospective analysis of 55 patients with EBV DNA-emia after allogeneic HSCT revealed an efficacy of prophylactic rituximab use. However, this study did not show a significant improvement of the overall survival and treatment-related mortality [101]. Rituximab use after allogeneic HSCT depletes both donor and recipient B-cells and delays B-cell immune reconstitution by at least 6 months [102]. Therefore, an early use of rituximab sometimes results in an increased incidence of critical cytopenia and infections [103,104]. Thus, the prophylactic use of rituximab should be limited as clinical trials or for patients at high risk of PTLD development. The prophylaxis by adoptive immunotherapy using EBV-specific cytotoxic T-cell for EBV-PTLD has also been reported, but the evidence and availability is limited [105].

### 6.2. Pre-Emptive Therapy

Pre-emptive therapy means an intervention for significant EBV-DNAemia in patients after HSCT who show no clinical manifestations of PTLD development. Rituximab is the sole recommended pre-emptive therapy for patients with EBV DNAemia after HSCT. However, the optimal clinical specimen in which to detect EBV-DNA (whole blood, plasma, serum, or peripheral blood mononuclear cells) has not been defined. In addition, the threshold to start pre-emptive therapy remains unclear. Some authors set a threshold of 1,000 copies/mL EBV DNA (detected by polymerase chain reaction), and reported that pre-emptive rituximab therapy reduced PTLD related mortality [98,100]. By contrast, a retrospective analysis of 332 adult patients with EBV DNAemia after HSCT revealed that pre-emptive rituximab therapy improved survival only in patients with ≥50,000 copies/mL EBV DNA [106]. The rate of increase of EBV copy number reflects the expansion of EBV-infected B-cells. Thus, a rapid increase of EBV-DNA is also considered as a trigger to start the pre-emptive therapy, although the cutoff value is not defined. Pre-emptive rituximab is usually administered at a dose of 375 mg/m^2^ once weekly with a total of 1–4 doses; this is based on the treatment response until the EBV-DNA load becomes negative [97]. A recent study evaluated low dose rituximab (100 mg/m^2^) as pre-emptive therapy and reported a good response which is comparable to the conventional therapy, although further evaluations are warranted [107].

### 6.3. Targeted Therapy

#### 6.3.1. Rituximab

Rituximab is used to treat PTLD after HSCT, as well as a pre-emptive therapy [108]. Rituximab is recommended as a first-line therapy for CD20-positive polymorphic or monomorphic PTLD after HSCT. It works by eliminating CD20-positive tumor cells and reducing the ratio of EBV-infected B-cells to EBV-specific T-cells, thereby favoring antiviral responses [109]. The initial response of PTLD patients to rituximab is estimated at 63–81%, with higher response rate being achieved when combined with RI, which also reduces the risk of GVHD [60,75,110]. Rituximab therapy is safe and well tolerated. However, the efficacy of rituximab is often lost if used for a long time because lymphoma cells downregulate expression of CD20 in response to the treatment. Therefore, the recommendation is that rituximab is administered once weekly for up to four doses. An additional concern is that rituximab use after allogeneic HSCT depletes both donor and recipient B-cells, thereby delaying B-cell immune reconstitution by at least 6 months [102]; this can result in an increased incidence of critical cytopenia and infection [103,104]. Rituximab is not effective against CD20-negative monomorphic PTLDs. In these cases, systemic chemotherapy based on each histological diagnosis would be selected as a first-line therapy. In addition, more advanced or refractory cases of CD20-positive PTLDs should first be treated with a combination of rituximab plus chemotherapy.

#### 6.3.2. Chemotherapy

Generally, immunochemotherapy is considered for patients who do not respond to RI and/or rituximab, or for those with specific histologic features such as T/NK cell lymphoma, Hodgkin’s lymphoma, Burkitt’s lymphoma, plasma cell neoplasms, primary CNS lymphoma, or other uncommon lymphoma subtypes. Information on the efficacy of chemotherapy obtained in the HSCT setting are limited; however, in general, data from the SOT setting suggest that patients with rare lymphoma subtypes should be treated with standard chemotherapy regimens for each specific histological feature, which have been demonstrated to improve the survival outcome of these patients [111,112]. However, patients with PTLD after allogeneic HSCT may carry the risk of further immunosuppression after systemic chemotherapy for PTLD, and also, they are more susceptible to chemotherapy-mediated toxicity because they have already received intensive conditioning regimen before HSCT. In addition, high rates of concomitant infection by bacteria, viruses, fungi, and parasites have been reported at the time of diagnosis of EBV reactivation or PTLD development [55]. Therefore, the treatment-related mortality in these situations is higher than that in immunocompetent patients with the same lymphoma subtypes; hence, chemotherapy is not recommended as a first-line treatment, except for these specific cases described above and for cases of late-onset EBV-negative PTLD [97].

#### 6.3.3. Adoptive Immunotherapy

Adoptive immunotherapy is performed by infusing patients with EBV-specific cytotoxic T-cells generated from serologically EBV-positive stem cell donors or third-party donors. The safety and efficacy of this treatment were first reported in studies involving its prophylactic use to prevent EBV reactivation and PTLD development in patients after allogeneic HSCT [113,114,115]. Because only EBV-specific cytotoxic T-cells are selected and used to induce cellular immunity to EBV-infected B-cells in the absence of GVHD, adoptive immunotherapy is very well tolerated; response rates are 46–85% when used to treat PTLD, although higher response rates (95%) are possible when using a sequential therapeutic strategy comprising a rituximab-based regimen followed by adoptive cellular immunotherapy [105,116,117,118]. Preparing donor-derived EBV-specific cytotoxic T-cells at the appropriate time is often difficult; therefore, banks of cryopreserved EBV-specific cytotoxic T-cells generated from third-party donors have been established in some countries [118,119]. However, applicability is still restrained due to several reasons including limited availability of donor cells and high costs.

#### 6.3.4. Possible Future Therapy

Recent pathological and molecular findings have led researchers to examine the therapeutic potential of several molecular targeting agents, including proteasome inhibitors, immunomodulatory agents, and PI3K inhibitors [120,121,122,123]. Most results are based on in vitro data, and further evaluation (in prospective clinical trials if possible) is necessary before such agents can be used as a treatment for patients with PTLD. As described above, EBV positivity is associated with copy number alterations and increased expression of PDL1 and PDL2 [31,124]. Immune checkpoint inhibitors have potential efficacy against EBV-positive PTLD by inducing T-cell immunity, and a phase II trial is ongoing (NCT03258567) [125]. Although there is no documentation in the literature, other CD20 antibodies, including ofatumumab or obinutuzumab, may also be effective for the treatment of PTLD, but they are more potent of an infusion reaction.

#### 6.3.5. Reduction of Immunosuppression

RI is defined as sustained decrease (at least 20%) in the dose of immunosuppressive drugs, regardless of the trough concentration [97]. Previously, the initial treatment for PTLD included RI alone to restore EBV-specific T-cell mediated immunity. However, RI is rarely effective for PTLD after HSCT when used alone. Moreover, graft rejection and GVHD development are constant concerns [126,127]. Therefore, RI alone is unsuitable for most PTLD cases developed after HSCT, and it must be combined with other strategies such as rituximab and/or chemotherapy.

#### 6.3.6. Other Strategy

Radiation therapy and surgical detection of tumors are also considered as a treatment for limited stage PTLD. The efficacy of antiviral drugs such as acyclovir, ganciclovir, foscarnet, and cidofovir, all advocated as treatments in the past, has not been demonstrated for EBV-PTLD; the recent general consensus is that these drugs are not useful for this disease [128,129].

#### 6.3.7. Management for Rare Cases

PTLDs with CNS involvement should be treated as primary CNS lymphoma. Combination therapies including high dose methotrexate and/or cytarabine, rituximab, intrathecal chemotherapy, IR, radiotherapy, and adoptive immunotherapy are treatments of choice [130,131]. However, an intensive chemotherapy is not tolerable for a part of patients after HSCT. According to a prospective study of 84 patients with EBV-PTLD, 6 of 10 patients with CNS involvement who had failed intravenous rituximab-based treatments achieved a complete response after intrathecal rituximab therapy [117]. Intrathecal rituximab is a possible therapeutic option, but its efficacy and safety have not been well evaluated. There are very rare cases of late-onset EBV-negative PTLD that develops 5 years after HSCT. As is clear from the genomic data discussed above, EBV-negative PTLD can also be regarded as malignant lymphoma coincidentally occurred in HSCT recipients, not as a genuine PTLD [97,132].

#### 6.3.8. Treatment Response Evaluation

Treatment response should be evaluated after initiation of any interventions. The goal of pre-emptive and targeted therapy is to reduce the EBV-DNA load, to improve clinical symptoms, and to achieve remission of the measurable lesions. Failure to respond to RI is usually defined when no improvement or progression of disease is noted after continuing RI for more than 2 to 4 weeks. Response to rituximab can be judged by a reduction of the EBV DNA load at least 1 log_10_ in the first week of treatment [97]. Risks for a poor response to rituximab are age 30 years or older, involvement of extra-lymphoid tissues, acute GVHD, and a lack of RI for PTLD [60]. Responses to targeted therapy are evaluated by PET-CT or CT in accordance with the Lugano criteria [95].

## 7. Prognosis

Although the introduction of rituximab and better supportive care has improved the outcome of patients with PTLD, the prognosis after development of PTLD in HSCT recipients is still poor when compared with that for diffuse large B-cell lymphoma in immunocompetent patients or with that for HSCT recipients without PTLD [61,75,132]. In addition, the prognosis of patients with PTLD developed after HSCT is worse than that developed after SOT. The 3 year overall survival of patients with PTLD following HSCT is 20–47%, whereas that of patients following SOT is 49–62% [6,60,61,62,118,133]. Generally, the condition of patients undergoing allogeneic HSCT is poor, principally due to intensive chemotherapy over a long period; also, the patients are profoundly immuno-compromised and may have been harboring infections prior to HSCT. In addition, there is always the risk of primary disease relapse; these factors may result in worse outcomes after HSCT. Several prognostic risk factors have been proposed. A large study evaluating the prognostic risk factors for PTLD after allogeneic HSCT identified age >30 years (hazard ratio (HR) = 2.2), malignant disease (HR = 2.6), no RI upon PTLD diagnosis (HR = 1.7), and acute GVHD grade II–IV at the time of PTLD diagnosis (HR = 3.2) as significant poor prognostic factors [60]. Other variables, such as poor performance status, elevated lactate dehydrogenase, CNS involvement, hypoalbuminemia, and response to rituximab, were reported as prognostic factors in PTLD patients after SOT; however, study results differ due to heterogeneity of disease, patient population, and treatments; and also, these risk factors have not been evaluated in the HSCT setting. A PTLD-1 trial suggested that the international prognostic index is a reliable prognostic marker for PTLD after SOT; however, this has not been validated in HSCT recipients [134,135].

Regarding the impact of EBV status on the prognosis of PTLD, although EBV-positive PTLD has a distinct genetic profiles from that of EBV-negative PTLD, EBV status does not affect the survival outcome of HSCT recipients with PTLD [88]. 

## 8. Conclusions

New insights into the biology of PTLD have led to development of new therapeutic options; however, the data of PTLD, particularly after HSCT, are limited, and no reliable treatment protocol has yet been established. Therefore, it is important to predict the risk of developing PTLD before undertaking HSCT. Prospective trials are urgently needed to establish the optimal treatment for each PTLD subtypes. Further, personalized treatments based on the genomic profile of individual patients are expected in the future.

## Figures and Tables

**Figure 1 cancers-12-00328-f001:**
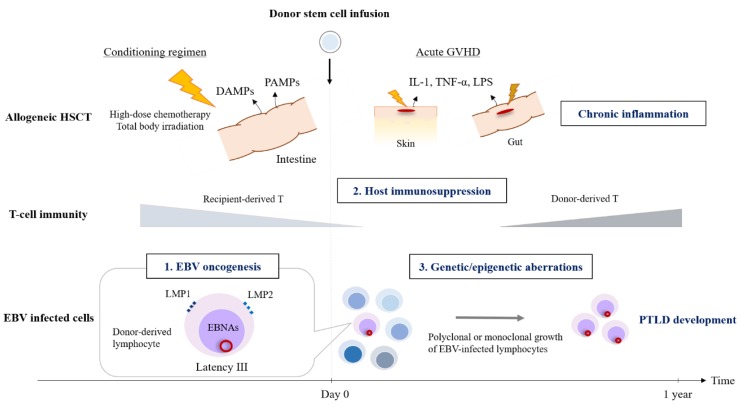
Pathogenesis of EBV-related PTLD after allogeneic stem cell transplantation. EBV-encoded oncogenes such as LMP 1 and LMP2; host immunosuppression due to the conditioning regimen; use of immunosuppressive agents; and growth advantages obtained by EBV-infected lymphocytes induced by genetic or epigenetic aberrations play an important role for development of PTLD. Persistent immune activation and chronic inflammation induced by the conditioning regimen and graft-versus-host disease also contribute to PTLD development. Abbreviations: HSCT, hematopoietic stem cell transplantation; GVHD, graft-versus-host disease; DAMPs, damage-associated molecular patterns; PAMPs, Pathogen-associated molecular patterns; IL-1, interleukin-1; TNF-αtumor necrosis factor-α; LPS, lipopolysaccharide; EBV, Epstein-Barr virus; LMP, latent membrane protein; EBNA, Epstein-Barr virus nuclear antigen; PTLD, post-transplant lymphoproliferative disorder.

**Figure 2 cancers-12-00328-f002:**
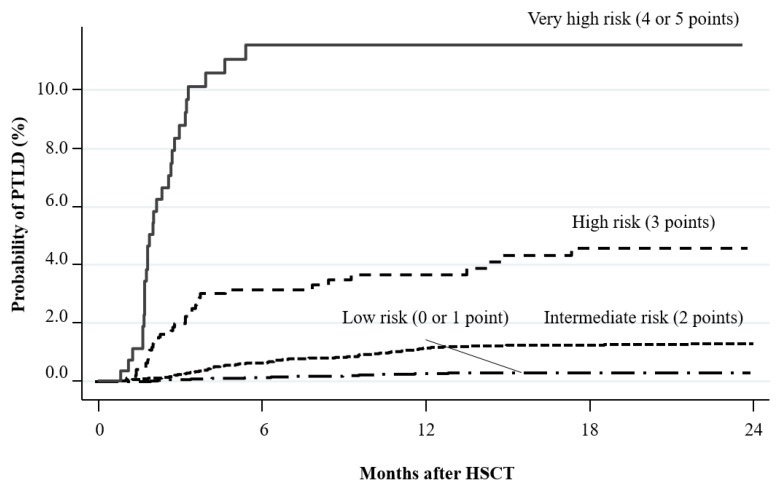
PTLD risk classification. Points are assigned to each risk factor: ATG use in the conditioning regimen (high dose, 2 points; low dose, 1 point), donor type (HLA-mismatched related donor, 1 point; unrelated donor, 1 point; cord blood, 2 points), and primary disease (aplastic anemia, 1 point). Based on the total number of points, the estimated incidence of PTLD at 2 years after HSCT is as follows: low risk (0–1 point), probability 0.3%; intermediate risk (2 points) probability 1.3%; high risk (3 points) probability 4.6%; very high risk (4–5 points) probability 11.5%. Abbreviations: PTLD, post-transplant lymphoproliferative disorder; HSCT, hematopoietic stem cell transplantation.

**Figure 3 cancers-12-00328-f003:**
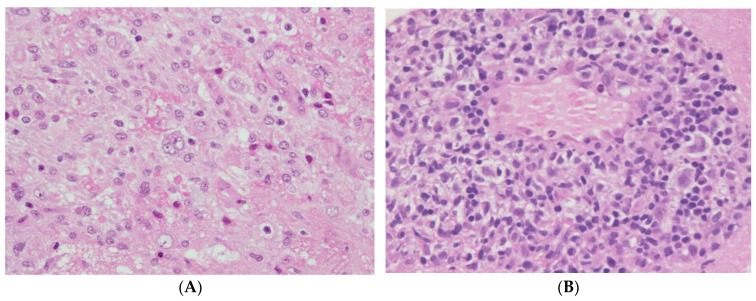
Hematoxylin and eosin staining of PTLD tissue samples from the central nervous system (×40 magnification). (**A**) Polymorphic PTLD. (**B**) Monomorphic PTLD. Abbreviations: PTLD, post-transplant lymphoproliferative disorder.

**Table 1 cancers-12-00328-t001:** Categories of PTLD and EBV status.

**1.**	**Non-destructive PTLDs**			**EBV status**
	1.1.	Plasmacytic hyperplasia		Almost 100% positive
	1.2.	Infectious mononucleosis		
	1.3.	Florid follicular hyperplasia		
**2.**	**Polymorphic PTLD**			>90% positive
**3.**	**Monomorphic PTLDs**			
	3.1.	B-cell neoplasms		Both EBV-positive and EBV-negative types exist (EBV-negative in 10–48% of cases, particularly T-cell lymphoma)
		Diffuse large B-cell lymphoma		
		Burkitt lymphoma		
		Plasma cell myeloma		
		Plasmacytoma		
		Other		
	3.2.	T-cell neoplasms		
		Peripheral T-cell lymphoma, NOS		
		Hepatosplenic T-cell lymphoma		
		Other		
**4.**	**Classical Hodgkin lymphoma PTLD**			>90% positive

Abbreviations: PTLD, post-transplant lymphoproliferative disorder; EBV, Epstein-Barr virus.

**Table 2 cancers-12-00328-t002:** Latency expression profiles of EBV infection.

Latency	EBV Proteins	Function of the Proteins	B-Cell Normal Counterpart	Post-Transplant Disease
**III (growth)**	EBER 1–2, **EBNA-LP**, **EBNA** 1–**2**, **EBNA 3A–C**, LMP 1, LMP 2A–B	Activate B-cells and promote growth and transformation of naïve B-cells	Activated B-lymphoblast	**PTLD**
**II (default)**	EBER 1–2, EBNA 1, **LMP 1–2A**	activate B-cells and differentiate naïve B-cells into memory B-cells through germinal center	Germinal center B-cell	(PTLD); Classical Hodgkin lymphoma; T/NK cell lymphoma
**I (EBNA1 only)**	EBER 1–2, **EBNA 1**	EBV genomic replication	Dividing memory B-cell	Burkitt lymphoma; Plasmablastic lymphoma
**0 (latency)**	EBER 1–2	Lifetime persistence of infection	Resting memory B-cell	Healthy carrier

Abbreviations: EBV, Epstein-Barr virus; EBER, Epstein-Barr virus-encoded small RNA; EBNA, Epstein-Barr virus nuclear antigen; LP, leader protein; LMP, latent membrane protein; PTLD, post-transplant lymphoproliferative disorder.

**Table 3 cancers-12-00328-t003:** Comparison of PTLD after HSCT with PTLD after SOT.

Variable	HSCT		SOT	
**Typical cell of origin**	Donor origin		Recipient origin	
**Frequency**	Cord blood	2.0–4.5%	Multi-visceral, small intestine	>20%
			Lung	3–10%
	Bone marrow or peripheral blood	0.8–4.0%	Heart	2–8%
			Kidney, pancreas, or liver	1–2%
**Onset time**	6–12 months		4–5.3 year	

Abbreviations: PTLD, post-transplant lymphoproliferative disorder; HSCT, hematopoietic stem cell transplantation; SOT, solid organ transplantation.

**Table 4 cancers-12-00328-t004:** Risk factors for PTLD following HSCT.

Variable	Category	Risk Factor	References
**Established risk factors**			
	T-cell depletion strategy	In vivo	[51,58,62,72]
		Ex vivo	[49,50,58]
	Donor	Unrelated BM/PBSC	[58,60]
		HLA-mismatched BM/PBSC	[58,60,61]
		Cord blood	[62]
**Other risk factors**			
1. Patient baseline	Disease	Aplastic anemia, primary immunodeficiency, chronic myeloid leukemia, advanced Hodgkin’s lymphoma	[53,62,70]
	Age	>50 years old	[58]
	Past medial history	Splenectomy	[61]
	Number of allogeneic HSCT	Two times or more	[62]
	EBV serological mismatch	EBV-negative recipient and EBV-positive donor	[54,61]
2. Factors before HSCT	Conditioning regimen	Reduced intensity conditioning	[61,79]
3. Factors after HSCT	Acute GVHD development	Grade II–IV	[58,61,62]
	MSC use		[61]
	CMV reactivation		[80]

Abbreviations: BM, bone marrow; PBSC, peripheral blood stem cell; HSCT, hematopoietic stem cell transplantation; EBV, Epstein-Barr virus; GVHD, graft-versus-host disease; MSC, mesenchymal stem cell; CMV, cytomegalovirus.

**Table 5 cancers-12-00328-t005:** Summary of the risk classification scoring systems.

Category	Risk Factor	Landgren, et al. [58] (CIBMTR/FHCRC)	Uhlin, et al. [61] (Karolinska Univ.)	Fujimoto, et al. [62] (JSHCT Database)
T-cell depletion	Selective T-cell depletion	●		
ATG use	GVHD prophylaxis	● *	●	● ^†^
	GVHD treatment			
Donor	HLA mismatch	●	●	● ^‡^
	Unrelated			
Age	50 years or older	●		
EBV status	Recipient −/donor +		●	
Conditioning regimen	Reduced intensity		●	
Acute GVHD II-IV			●	
Splenectomy			●	
MSC treatment			●	
Disease	Aplastic anemia			●

●: These factors are components of each risk classification. * Only two HLA antigen mismatched siblings or unrelated donors, accompanied by selective T-cell depletion methods or ATG therapy were included; ^†^ High dose ATG was assigned 2 points, whereas low dose ATG was assigned 1 point. ^‡^ Cord blood was assigned 2 points, and the others were assigned 1 point. Abbreviations: ATG, antithymocyte globulin; EBV, Epstein-Barr virus; GVHD, graft-versus-host disease; MSC, mesenchymal stem cell; CIBMTR, Center for International Blood and Marrow Transplant Research; FHCRC, Fred Hutchinson Cancer Research Center; JSHCT, Japan Society for Hematopoietic Cell Transplantation.

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
