# Peer review of "Epstein-Barr Virus-Associated Post-Transplant Lymphoproliferative Disorders after Hematopoietic Stem Cell Transplantation: Pathogenesis, Risk Factors and Clinical Outcomes"

_cancers, 2020, doi:10.3390/cancers12020328_

Round 1
Reviewer 1 Report
This manuscript reviews EBV-associated PTLD which is an important issue after allogeneic hematopoietic stem cells or solid organ transplantation. Focuses of this manuscript includes pathogenesis, epidemiology, risk factors, treatments, and prognosis of EBV-associated PTLD. This manuscript is a well-written and comprehensive review article. I have no further comments.
Author Response
Thank you for your comments.
Reviewer 2 Report
The authors review the pathogenesis, risk factors and clinical outcomes for Epstein-Barr virus (EBV) associated posttransplant lymphoproliferative disorder. This is an important field especially in the field of allogeneic stem cell transplantation. I would like to see the authors to focus only on EBV associated disease in this patient group, and the title to reflect such a focus. There are already some published reviews from 2-4 years ago, and I would like the authors to write THE review of EBV associated lymphoproliferative disorders in allotransplant stem cell recipients where both pathologists as well as clinicians (hematology, infectious diseases) can find what they need. This would require a much more detailed and extended review.
Major comments:
I would like to see pictures to illustrate the various histological subtypes of this disease. A much more detailed discussion of posttransplant EBV viremia (i.e. preemptive treatment) should be included, including the clinical handling (e.g. response evaluation, practical guidelines) A balanced and more detailed discussion of the prognostic model (Figure 2) should be included. What are the alternatives for prognostic evaluation of these patients. A much more detailed discussion of the treatment, the practical guidelines and how to evaluate responses should be included. The questions of grading and of endorgan disease should be addressed, including CNS disease.
Minor comments:
What is late-onset EBV negative PTLD? How late is late onset? Is prophylactic treatment an alternative?
Author Response
Reply to reviewer #2
Major comments
As suggested by the reviewer, we have included histological images of H&E-stained CNS biopsy taken from a patient with monomorphic PTLD and form a patient with polymorphic PTLD (Figure 3).
As suggested by the reviewer, we have discussed preemptive therapy, response evaluation, and practical guidelines for PTLD. These have been included in section “6. Treatments”. We also discuss CNS involvement of PTLD in this section. In addition, we discussed how to evaluate PTLD stage (see section “5. Clinical presentation”).
As suggested, we have discussed other risk classification models (in addition to Figure 2), and have included a new “Table 5”, which compares these classification models.
Minor comments
Late-onset EBV-negative PTLDs can occur after both HSCT and after SOT. These are less common in HSCT recipients and the etiology is uncertain. However, we think that it is important to refer to such cases, so included these in section “5. Clinical presentation” and “6. Treatments”. The definition of “late” onset depends on the reports, but the ECIL-6 guideline defined it as 5 years. Therefore, in this review, we described late-onset PTLD as PTLD developing more than “5 years” after HSCT (see section “6. Treatments”).
The efficacy and recommendations for PTLD prophylaxis are not yet established. We added this statement in section “6. Treatments”.

Reviewer 3 Report
Fujimoto et al. give a nice overview on EBV+ PTLD after HSCT covering different aspects of the disease such as pathogenesis, epidemiology, risk factors and therapy. I have only a few comments:
1) The title of the paper does not reflect the fact that it is only dealing with PTLD after HSCT,
2) Authors may also refer to PD-L1 and PD-1 expression in EBV+ PTLD and their potential role for therapy with check point inhibitors.
3) Authors may point more clearly to the differences in PTLD after SOT and HSCT,
4) Authors may include a chapter on clinical presentation of patients with PTLD after HSCT,
Author Response
Reply to reviewer #3
Thank you for these comments.
As suggested by the reviewer, we added “after hematopoietic stem cell transplantation” to the title (page 1).
We agree that the expression of PD-L1 in EBV+ PTLD and the role of checkpoint inhibitor for therapy are important issue. Therefore, we have added the statements for these in the section “6. Treatments”.
As suggested, we added a new “Table 3” to clearly show the differences between PTLD after HSCT and PTLD after SOT.
We have generated a new section, entitled “5. Clinical presentation” in this manuscript.

Round 2
Reviewer 2 Report
The Authors have prepared a very extensive review, and they have addressed all my comments. I think the review now covers what we need, an extensive and updated and complete review. I have some minor suggestion that the Authors should address, and it should also be easy for them to do so.
Comments:
Most important, what would the authors do if they have a patient With severe side effects to rituximab? Has any kind of evidence been published with respect to other alternative B cell antibodies? Please address/comment this question a bit more, but still briefly. This is a question of clinical relevance. Please do a careful control of the language in the new parts of the manuscript, e.g. lines 301-302, 395, 397. Please redesign Table 3 so that Frequency, 0.8-4.0% and Depends on the type.... comes on the same line/same Level. Table 5: please use black/filled symbols and not open rings to make it easier to see the differences for the reader. Figure text 2: I think abberviations should be a part of the figure legend and not be inserted between the figure and the legend. Lines 361-362: Does the figure legend have the correct size of the letters (see for example legend to Figure 2).
Author Response
Thank you for these comments.
We believe that rituximab is almost always safe and applicable (page 12, line 412). Theoretically, other CD20 antibodies including ofatumumab or obinutuzumab may also be effective for the treatment of PTLD, but they are more potent of infusion reactions. Side effects of rituximab other than the infusion reaction are rare or usually tolerable. So, the alternative role of those antibodies is limited in the context of safety. Much interest will be paid for their anti-tumor role in future. We have added these statements in page 13, lines 455-456. To organize the treatments section clearly, we have also added subheadings to “6.3. Targeted therapy”.
Thank you for comment. We will further undergo the English editing service of this journal after completing this revision.
We have redesigned the Table 3 so that the frequency comes on the same column. Also, we have added the sentences about the incidence after cord blood transplantation (page 7, lines 215-217).
According to the reviewer’s suggestion, we have revised the Table 5.
As suggested by the reviewer, we have moved the position of abbreviations in Figures 1 and 2.
